# GeoLS: Geodesic Label Smoothing for Image Segmentation

**Sukesh Adiga Vasudeva**      SUKESH.ADIGA-VASUDEVA.1@ENS.ETSMTL.CA
**Jose Dolz**      JOSE.DOLZ@@ETSMTL.CA
**Herve Lombaert**      HERVE.LOMBAERT@ETSMTL.CA
*ETS Montreal, Canada*

**Editors:** Accepted for publication at MIDL 2023

## Abstract

Smoothing hard-label assignments have emerged as a popular strategy in training discriminative models. Nevertheless, most existing approaches are typically designed for classification tasks, ignoring underlying properties of dense prediction problems, such as medical image segmentation. First, these strategies often ignore the spatial relations between a given pixel and its neighbours. And second, the image context associated with each label is overlooked, which can convey important information about potential errors or ambiguities in the segmentation masks. To address these limitations, we propose in this work geodesic label smoothing (GeoLS), which integrates image information into the label smoothing process by leveraging the geodesic distance transform of the images. As the resulting label assignment is based on the computed geodesic map, class-wise relationships in the soft-labels are better modeled, as it considers image gradients at the boundary of two or more categories. Furthermore, spatial pixel-wise relationships are captured in the geodesic distance transform, integrating richer information than resorting to the Euclidean distance between pixels. We evaluate our method on two publicly available segmentation benchmarks and compare them to popular segmentation loss functions that directly modify the standard hard-label assignments. The proposed geodesic label smoothing improves the segmentation accuracy over existing soft-labeling strategies, demonstrating the validity of integrating image information into the label smoothing process. The code to reproduce our results is available at: https://github.com/adigasu/GeoLS

**Keywords:** Image Segmentation, Geodesic Distance, Label Smoothing

## 1. Introduction

Deep learning is driving progress in solving complex predictive tasks across a wide range of visual recognition problems, including medical image segmentation (Litjens et al., 2017; Hesamian et al., 2019). A common strategy to train these models is to optimize a cross-entropy objective function, which minimizes the differences between the predicted posterior probabilities and the ground-truth distributions. Nevertheless, while this learning objective has demonstrated high performance in independent class prediction tasks such as image classification, its use in dense prediction problems might be suboptimal. Indeed, image segmentation is a massively structured and dense problem, as class predictions at each pixel are inherently conditioned to the spatial relationship with surrounding regions. Furthermore, inter-class relation is typically overlooked because the ground-truth objective is modeled as a one-hot encoding vector. However, capturing this relation is extremely important in medical image segmentation due to the ambiguity in the boundaries between neighboring

regions, which might lead to imprecise boundary annotation. Thus, novel learning objectives that explicitly model these pixel-wise and class-wise relationships would improve the training of segmentation models.

A popular strategy to model inter-class relations consists in modifying the hard-label assignments so that the ground-truth annotations used in training become a soft version of the original one-hot labels. For example, label smoothing (LS) (Szegedy et al., 2016) generates a new soft-label assignment by reducing the weight of the target class, which is redistributed uniformly over all the other classes. To capture the underlying structure on class labels, where the distance between categories is important, (Galdran et al., 2020) proposed a non-uniform LS. They replace the uniform distribution over non-target classes with a Gaussian distribution centered at the target class. The benefit of LS has been used for medical image classification (He et al., 2020; Islam et al., 2020). Smoothing of a target label can also be achieved based on model prediction. For instance, Focal loss (FL) (Lin et al., 2017) down-weights the target labels for well-predicted samples and up-weights them for the mis-classified samples. FL has shown potential to improve the performance of independent class predictions (Tran et al., 2019; Ahmed et al., 2022) and dense prediction tasks (Abulnaga and Rubin, 2019). However, FL does not explicitly consider pixel relationships, which is fundamental in the context of image segmentation.

Soft-labeling can also be based on dilated regions of target masks (Kats et al., 2019), adding granularity in ambiguous object boundaries. Further structural labeling ambiguity is modeled using a spatially varying label smoothing (SVLS) (Islam and Glocker, 2021). Their soft-label probabilities are based on spatial variations of pixels within Gaussian-smoothed label distributions from target masks. These approaches solely revolve around the provided target labels, which may be unreliable. For instance, image boundaries between anatomical regions may be ambiguous and poorly defined due to imaging or existing pathologies, being prone to label errors (Joskowicz et al., 2019). Explicitly modeling the ambiguity of image boundaries should therefore be considered when training a segmentation model.

These limitations motivate our approach, which leverages the geodesic distance transform to model inter-pixel and inter-class relationships. We argue that including geodesic maps in the segmentation model has the potential to improve accuracy, as they model distances based on pixel content across objects (Toivanen, 1996; Criminisi et al., 2008). We, therefore, propose a novel Geodesic Label Smoothing (GeoLS) for the segmentation of medical images. Specifically, we integrate the geodesic maps obtained for each target object to (i) capture spatial information between the other objects and (ii) capture image uncertainty in the form of soft-labels. In contrast to existing soft-labeling approaches, our GeoLS smooths the label using geodesic maps, which capture spatial relations and image context needed for medical image segmentation. Our method is extensively validated on two different medical imagining datasets: the 2019 brain tumour segmentation (BraTS) challenge dataset (Bakas et al., 2017, 2018) and the 2021 Abdominal organ segmentation dataset (Ma et al., 2022). The results demonstrate the superiority of our approach over the state-of-the-art methods on soft-label segmentation.

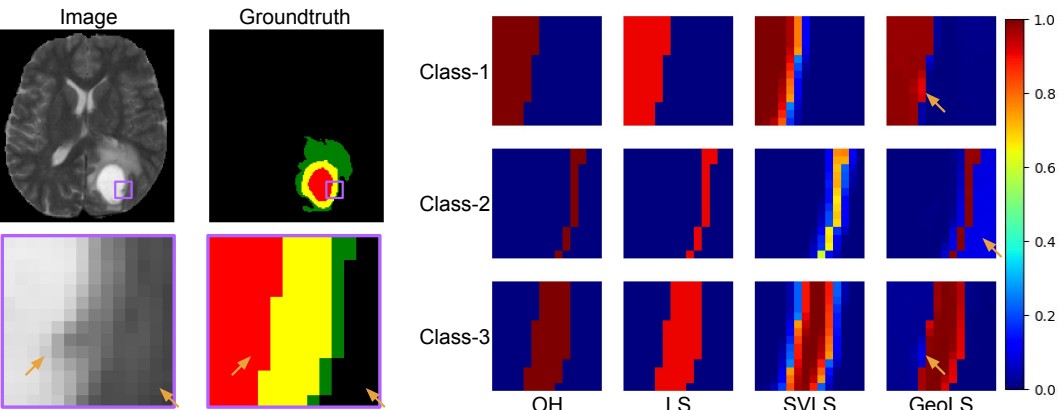

Figure 1: Comparison of soft-labeling from One-Hot (OH) with Label Smoothing (LS), Spatially Varying LS (SVLS), and ours (GeoLS) on a brain tumour. Left side: An image, its ground-truth mask, and an image closeup with ground-truth around a tumour. Right side: The class probabilities (%) for each method (OH, LS, SVLS, GeoLS) in the same closeup area. OH/LS ignores inter-class spatial relations, whereas SVLS redistributes probabilities to neighboring classes but simply blurs OH labels. GeoLS redistributes probabilities based on image gradients from geodesic maps, thus better capturing ambiguous boundaries (orange arrows).

## 2. Method

### 2.1. Preliminaries

Let us denote $D = \{(x_i, y_i)\}_{i=1}^N$ as the training dataset, where $x_i \in \mathbb{R}^{S \times H \times W}$ represents an input volume and $y_i \in \{0, 1\}^{C \times S \times H \times W}$ its associated segmentation mask, modeled as one-hot encoding, with $C$ being the number of classes. For a given volume, the segmentation network is typically trained using the standard cross-entropy loss function as:

$$\mathcal{L}_{CE} = -\sum_{c=1}^{C} \sum_v y_{c,v} \log(p_{c,v}), \tag{1}$$

where $p_{c,v}$ is the predicted softmax probability of the segmentation network, and $v$ is a voxel. For simplicity, we drop $i$ and $v$ notations and use them wherever necessary, and assume the loss function is normalized by the cardinality of the training set.

The hard-label assignments, $y_i$, do not capture inter-class relations, which fail to provide the model with annotation ambiguity. To model the inter-class relation, a simple strategy consists in softening the hard-labels during training. Label smoothing (LS) (Szegedy et al., 2016) generates a soft-label by reducing the target class probability with a factor $\alpha$ and redistributes it uniformly to all other classes:

$$y_c^{LS} = (1 - \alpha)y_c + \frac{\alpha}{C} \tag{2}$$

Training a segmentation network with LS is carried out by replacing the hard-label $y_i$ with the soft-label $y_i^{LS}$.

## 2.2. Geodesic Distance Transform.

Segmentation is a dense pixel-wise classification problem, which depends not only on the spatial relation of neighboring pixels but also on the image intensities. For instance, pixels with varying intensities might have the same class label in the inner region of the target object than in its boundaries. However, such ambiguity near the object boundaries is not captured in the standard labels. In order to integrate these intensity differences in the training of segmentation networks, we integrate the geodesic distance transform (Toivanen, 1996), which captures the variation in the image gradients as a function of the distance.

Suppose $\mathcal{S}$ represents the set of seed voxels belonging to target class regions. The unsigned geodesic distance map of each voxel $v$ to the set $\mathcal{S}$ of each class is defined as:

$$D_c(v; \mathcal{S}, x_i) = \min_{v' \in \mathcal{S}} d(v, v'), \tag{3}$$

with:

$$d(v, v') = \min_{p \in P_{v,v'}} \int \sqrt{||p'(s)||^2 + \gamma^2 (\nabla x_i \cdot u(s))^2} ds, \tag{4}$$

where $P_{v,v'}$ is the set of all paths between voxels $v$ and $v'$, and $p$ being one of such feasible paths, parameterized by $s \in [0, 1]$. Furthermore, a unit vector is defined as $u(s) = \frac{p'(s)}{||p'(s)||}$, which is tangent in the direction of the path. The term $\gamma$ controls the contribution of the image gradient ($\nabla x_i$) versus the spatial distances to the set $\mathcal{S}$. Note that, Eq. 4 reduces to the Euclidean Distance for $\gamma = 0$ and to compute the geodesic distance $\gamma$ is set to 1 (Criminisi et al., 2008). Finally, we obtain the geodesic map for each target class as

$$g_c = e^{-D_c}, \tag{5}$$

which are subsequently normalized to range $[0, 1]$.

## 2.3. Geodesic Label Smoothing (GeoLS)

The geodesic maps capture the image gradient information, which can be useful in smoothing the hard-labels and yet exploit image context. Instead of adding the smoothness uniformly as in LS (Szegedy et al., 2016), or smooth with a Gaussian filter such as in SVLS (Islam and Glocker, 2021), our approach smooth the label availing geodesic maps. This will integrate image context into the target label assignment. To achieve this, we first normalize the geodesic map for each class as $\tilde{g}_c = \frac{g_c}{\sum_c g_c}$, such that it follows a probability distribution. The geodesic probability distribution is consequently combined with the original target probability to generate the soft-label, as follows:

$$y_c^{GeoLS} = (1 - \alpha)y_c + \alpha\tilde{g}_c \tag{6}$$

These generated geodesic soft-label $y_c^{GeoLS}$, is thereupon plugged in Eq. 1 to train the segmentation network. Figure 1 demonstrates the comparison of GeoLS over one-hot (OH), LS, and SVLS labels. The class probabilities in LS were reduced compared to OH without considering the spatial structure. SVLS considers the spatial context based on the blurring of OH labels, whereas GeoLS captures both the local spatial variation and the image context.

## 3. Experiments

### 3.1. Datasets and Evaluation Metrics

To validate our GeoLS, we use two publicly available benchmarks: a) the Brain Tumour Segmentation dataset from the BraTS 2019 challenge (Bakas et al., 2017, 2018) and b) the abdomen multi-organ segmentation dataset from the FLARE challenge (Ma et al., 2022). The description of these datasets and how they are used in our experimental setting is detailed below.

**BraTS 2019.** It includes 335 multimodal brain MRI scans and corresponding glioma tumour masks. Each scan has MR sequences of FLAIR, T1, T1ce, and T2, which are pre-processed with skull-stripping, co-registred to a fixed template and resampled to an isotropic resolution of 1 $mm^3$. The annotation consists of the necrotic and non-enhancing core, edema, and enhancing tumour regions, which are mapped to Whole Tumour (WT), Tumour Core (TC), and Enhancing Tumour (ET) during evaluation. For all our experiments, a fixed dataset split of 235 for training, 32 for validation, and the remaining 68 for testing.

**FLARE 2021.** The dataset comprises 361 abdominal CT scans with corresponding segmentation masks of four organs: liver, kidney, spleen, and pancreas. These scans have varying resolutions, which are first resampled to a uniform resolution of $2 \times 2 \times 2.5$ $mm^3$ and then normalized by clipping the intensity values outside the $[0.5, 0.95]$ percentile range. We use a fixed dataset split of 260 for training, 26 for validation, and the remaining 75 for testing for all our experiments.

**Evaluation Metrics.** We employ commonly used segmentation metrics, namely Dice Score Coefficient (DSC), Surface Dice (SD) (Nikolov et al., 2018), and 95% Hausdorff Distance (HD) to evaluate the discriminative performance. All experiments are run three times with a fixed set of seeds on the same machine, and their average results are reported.

### 3.2. Training and implementation details.

To assess the contribution of our GeoLS, a 3D U-net (Çiçek et al., 2016) architecture is used as the segmentation model in all our experiments. The model is trained using an Adam optimizer (Kingma and Ba, 2015) with a learning rate of 1e-4 and weight decay of 1e-4. The input of the segmentation network is center-cropped to $128 \times 192 \times 192$ in BraTS and $112 \times 160 \times 208$ in FLARE. We employ online data augmentation, including random flipping and rotation of input images, as in (Islam and Glocker, 2021). The network is trained for 200 epochs with a batch size of 4, and the best validation model is used for testing. Our evaluation includes experiments with cross-entropy (CE), LS (Szegedy et al., 2016), FL (Lin et al., 2017), and SVLS (Islam and Glocker, 2021) losses as training objectives. For both LS and our model, the smoothing factor $\alpha$ is set to 0.1 (Müller et al., 2019). An open-source library (GeodisTK) is used to generate the geodesic maps with target label skeleton as seed points $\mathcal{S}$. All experiments were run on an NVIDIA RTX A6000 GPU with PyTorch 1.8.0.

Table 1: **Segmentation results on the BraTS test set.** The best and second-best results are highlighted in bold and underlined for each tumour structure (ET, TC, WT).

| | Methods | ET | TC | WT | Average |
|---|---|---|---|---|---|
| **DSC (%) ←** | CE | 72.05 ± 2.14 | 82.38 ± 0.91 | 90.09 ± 0.39 | 81.51 ± 1.03 |
| | LS | **73.28 ± 0.85** | 82.65 ± 0.30 | **90.46 ± 0.08** | 82.13 ± 0.35 |
| | FL | 72.86 ± 0.68 | 82.74 ± 0.13 | 89.95 ± 0.73 | 81.85 ± 0.36 |
| | SVLS | 73.15 ± 2.82 | 82.67 ± 1.96 | 90.43 ± 0.78 | 82.08 ± 1.81 |
| | GeoLS (ours) | 72.98 ± 1.27 | **83.36 ± 0.97** | **90.46 ± 0.25** | **82.27 ± 0.77** |
| **SD (%) ←** | CE | 81.35 ± 2.91 | 80.92 ± 1.34 | 90.09 ± 2.07 | 84.12 ± 1.77 |
| | LS | 82.04 ± 2.58 | 80.86 ± 1.55 | 93.08 ± 0.54 | 85.33 ± 1.55 |
| | FL | 82.55 ± 1.51 | 80.79 ± 0.97 | 92.52 ± 0.26 | 85.29 ± 0.86 |
| | SVLS | 83.71 ± 2.83 | 80.70 ± 2.39 | 92.47 ± 1.37 | 85.63 ± 2.02 |
| | GeoLS (ours) | **84.26 ± 2.77** | **82.86 ± 0.29** | **93.37 ± 0.71** | **86.83 ± 1.15** |
| **HD (mm) →** | CE | 14.55 ± 1.61 | 7.64 ± 1.15 | 6.28 ± 0.86 | 9.49 ± 1.20 |
| | LS | 13.52 ± 0.35 | 7.23 ± 0.16 | 5.95 ± 0.16 | 8.90 ± 0.21 |
| | FL | 13.53 ± 1.29 | 6.90 ± 1.26 | 6.08 ± 0.88 | 8.83 ± 1.10 |
| | SVLS | **12.83 ± 2.70** | 6.93 ± 1.37 | **5.72 ± 1.10** | **8.50 ± 1.70** |
| | GeoLS (ours) | 13.71 ± 1.54 | **6.72 ± 1.64** | 5.90 ± 0.76 | 8.78 ± 1.28 |

Table 2: **Segmentation results on the FLARE test set.** The best and second-best results are highlighted in bold and underlined.

| | Methods | Liver | Kidney | Spleen | Pancreas | Average |
|---|---|---|---|---|---|---|
| **DSC (%) ←** | CE | 94.88 ± 0.31 | 94.70 ± 0.33 | 95.46 ± 0.85 | 72.52 ± 0.61 | 89.39 ± 0.14 |
| | LS | **95.96 ± 1.11** | **94.89 ± 0.35** | 95.61 ± 0.63 | 73.07 ± 1.35 | 89.88 ± 0.38 |
| | FL | 94.78 ± 0.68 | 94.13 ± 0.44 | 93.84 ± 1.01 | 67.22 ± 1.31 | 87.49 ± 0.32 |
| | SVLS | 95.12 ± 1.46 | 94.30 ± 0.28 | 95.12 ± 0.31 | 70.49 ± 2.52 | 88.76 ± 1.00 |
| | GeoLS (ours) | 95.60 ± 0.87 | 94.80 ± 0.37 | **96.52 ± 0.30** | **73.72 ± 1.02** | **90.16 ± 0.44** |
| **SD (%) ←** | CE | 90.96 ± 0.53 | 94.08 ± 0.40 | 93.62 ± 0.55 | 69.00 ± 0.96 | 86.91 ± 0.11 |
| | LS | 91.49 ± 0.45 | 95.00 ± 0.13 | **94.57 ± 0.66** | **70.67 ± 1.01** | **87.94 ± 0.08** |
| | FL | 89.28 ± 0.30 | 92.66 ± 1.39 | 93.50 ± 0.50 | 61.65 ± 1.19 | 84.27 ± 0.61 |
| | SVLS | 90.88 ± 0.72 | **95.55 ± 0.30** | 92.82 ± 1.80 | 66.94 ± 1.76 | 86.55 ± 1.08 |
| | GeoLS (ours) | **91.07 ± 0.61** | 94.33 ± 1.20 | 94.47 ± 0.82 | 69.69 ± 1.22 | 87.39 ± 0.25 |
| **HD (mm) →** | CE | 4.15 ± 1.10 | 2.94 ± 0.11 | 2.98 ± 1.06 | 6.72 ± 1.18 | 4.20 ± 0.19 |
| | LS | **2.87 ± 1.14** | 2.93 ± 0.37 | 2.60 ± 0.24 | 6.37 ± 1.03 | 3.69 ± 0.26 |
| | FL | 3.76 ± 1.19 | 3.36 ± 0.90 | 3.55 ± 1.30 | 7.84 ± 1.31 | 4.63 ± 0.65 |
| | SVLS | 4.31 ± 1.39 | 3.09 ± 0.41 | 3.51 ± 0.69 | 8.15 ± 1.81 | 4.77 ± 1.00 |
| | GeoLS (ours) | 3.01 ± 1.05 | **2.40 ± 0.5** | **1.49 ± 0.55** | **5.59 ± 0.20** | **3.12 ± 0.21** |

## 4. Results

The performance of the proposed geodesic label smoothing approach is compared with existing soft-label approaches and report their discriminative results in Tables 1 and 2. We first validate our GeoLS on the multiclass brain tumour segmentation and report both individual tumour and average results in Table 1. From the results, we can observe that employing soft-labels improves the performance across all models and metrics. Among soft-label methods, our method outperforms LS and SVLS in terms of DSC and SD scores, respectively,

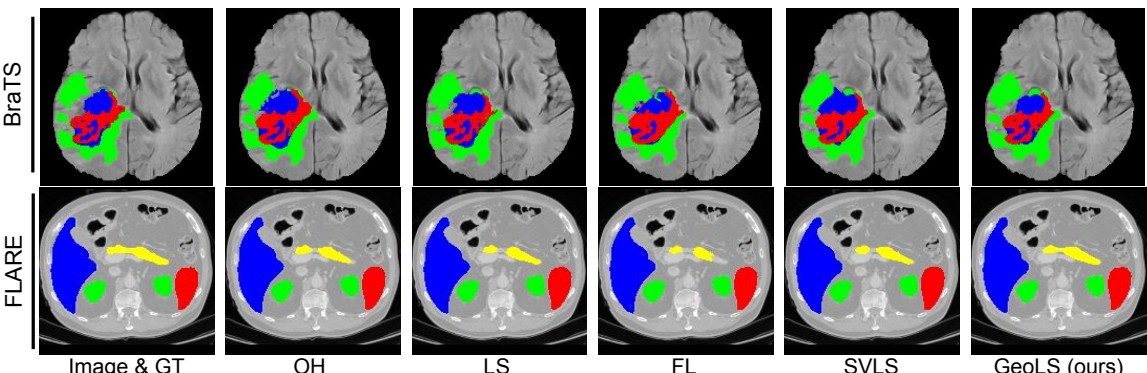

Figure 2: **Qualitative comparison on BraTS (top) and FLARE (bottom) datasets.** Coloring indicates different tumour structures on top and organs on bottom. GeoLS shows improvements in ambiguous boundaries, such as the pancreas (yellow).

and is competitive with SVLS in HD metric. Notably, our method yields improvements in the SD score (1.2%), highlighting the improvement in the boundary regions.

Furthermore, the results of multi-organ abdomen segmentation on the FLARE dataset are reported in Table 2. We can see a similar trend in LS and GeoLS results. In contrast, a noticeable performance gap is observed in FL and SVLS compared to CE results, possibly due to the over-suppression of one-hot labels in the boundaries. Moreover, existing methods are ranked differently across metrics and datasets, indicating that these methods are sensitive to datasets. Our method invariably outperforms the existing approach in most cases. From these results, our method is consistent across different datasets, emphasizing the robustness of geodesic soft-labels.

**Qualitative Results.** Visual segmentation results of brain tumour from BraTS and abdominal organs from FLARE by different methods are depicted in Fig. 2. In the top row of the figure, the predictions of existing methods are predominantly over-segmenting in enhancing tumour (red) (OH, FL, SVLS) and necrotic and non-enhancing core (blue) (OH, LS, FL, SVLS). In contrast, our method minimizes misclassification errors and produces a superior segmentation. In multi-organ segmentation (bottom), the challenging pancreas region (yellow) is predominantly under-segmented by all existing methods. Our method produces a better segmentation in the pancreas region than existing methods, which is supported by the quantitative results provided in previous sections. We argue that this improvement may be due to the inclusion of image gradient information from geodesic maps.

**Smoothing Factor $\alpha$.** Figure 3 shows the sensitivity of the smoothing factor $\alpha$ (in Eq.6) versus segmentation performance. Specifically, we assess the segmentation performance using DSC and HD scores by varying the $\alpha$ values on both BraTS and FLARE datasets. Note that $\alpha = 0$ is equivalent to CE loss, which does not leverage geodesic maps. Results demonstrate that increasing alpha values show a gradual segmentation performance improvement in both DSC and HD scores for FLARE, and the best result is obtained for $\alpha = 0.1$. In BraTS, segmentation accuracy is almost flat for the DSC score for initial values of $\alpha$, whereas the HD score improves slightly. The best scores are obtained for $\alpha = 0.05$.

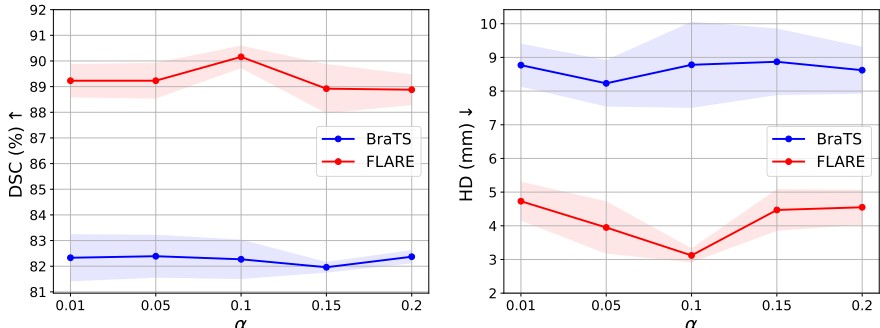

Figure 3: **Sensitivity of smoothing factor** $\alpha$ - Each point in the line indicates the DSC (left) and HD (right) scores on BraTS and FLARE datasets. The segmentation performance on both scores and datasets is consistent for $\alpha = 0.1$.

Beyond $\alpha = 0.1$, the performance generally decreases for both datasets. We choose $\alpha = 0.1$ for all experiments since it is more consistent in both scores and datasets.

Table 3: **Performance under different seed sets** $\mathcal{S}$ - Average DSC and HD scores on BraTS and FLARE are reported. The best and second-best results are highlighted in bold and underlined.

| Datasets | **BraTS** | | **FLARE** | |
|---|---|---|---|---|
| **choice of $\mathcal{S}$** | **DSC (%)** ↑ | **HD (mm)** ↓ | **DSC (%)** ↑ | **HD (mm)** ↓ |
| random-3 | **82.98 ± 0.68** | **8.10 ± 0.09** | 87.83 ± 1.02 | 4.79 ± 0.16 |
| random-5 | 82.51 ± 0.80 | 9.00 ± 0.70 | 89.46 ± 1.00 | 4.20 ± 0.97 |
| random-7 | 82.36 ± 0.48 | 8.89 ± 0.81 | 89.23 ± 0.21 | 4.41 ± 0.49 |
| **skeleton** | 82.27 ± 0.77 | 8.78 ± 1.28 | **90.16 ± 0.44** | **3.12 ± 0.21** |
| erosion | 81.93 ± 0.93 | 9.17 ± 0.68 | 89.56 ± 0.08 | 3.63 ± 0.27 |

**Choice of seed set $\mathcal{S}$.** The geodesic map varies with the choice of seed set $\mathcal{S}$, as the distance transform is calculated for each pixel in the image to the seed points in $\mathcal{S}$. Therefore, we evaluate the segmentation accuracy with varying choices of seed-set strategies. To generate different geodesic maps, seed sets are generated using a random selection of 3, 5, and 7 pixels inside each target class as seed points. In addition to random generation, seed sets are generated using remainings of skeletonization and an erosion operation of each target class. Table 3 reports all comparable segmentation performances, indicating the robustness of our geodesic maps to varying choices of seed points. The results may also indicate that the skeleton-based seed points could be further explored as a viable seeding strategy.

## 5. Discussion and Conclusion

Existing hard and soft-labeling approach for image segmentation ignores the spatial relation and image context embedded in the task. Therefore, explicitly modeling of such information would improve the training of segmentation networks. This work proposes a Geodesic label smoothing (GeoLS) for the segmentation of medical images. It yields soft-labels by including the image content in the label smoothing process via the geodesic distance transform.

Leveraging geodesic soft-labels in model training improves the segmentation performance. Results demonstrate that the proposed method achieves consistent performance compared to the existing soft-label approach on multiclass brain tumour and abdomen segmentation from 3D MR and CT volumes, respectively. The proposed Geodesic label smoothing is orthogonal to other types of segmentation losses, such as Dice loss. It can be combined to form a compound loss, such as a combination of CE and Dice. However, this work aims to provide an alternative to hard and soft-labeling losses. Furthermore, our geodesic soft-label can be adapted to a broader range of applications where supplementing image content is beneficial.

## Acknowledgments

This research work was partly funded by the Canada Research Chair on Shape Analysis in Medical Imaging, the Natural Sciences and Engineering Research Council of Canada (NSERC), and the Fonds de Recherche du Quebec (FRQNT). We also thank Mustafa Chasmai and Kumar Devesh for their discussions and insights on this work.

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
