# OpenReview forum: "GeoLS: Geodesic Label Smoothing for Image Segmentation"
_MIDL.io/2023/Conference — MIDL 2023 Oral_

### Official Review · Reviewer_kqR3 · 2023-01-29

**Confidence:** 5
**Preliminary Rating:** 5
**Recommendation:** Poster

**Summary:**

The key idea of this paper is to propose a new label smoothing technique, called geodesic label smoothing (GeoLS), which integrates image information into the process. This method considers both class-wise relationships between labels and spatial pixel-wise relationships captured in the geodesic distance transform. To evaluate their proposed method, they conducted experiments on two publicly available segmentation benchmarks and compared it with existing soft-labelling strategies. The results showed that using GeoLS improved the segmentation accuracy over other methods, demonstrating its validity for integrating image information into label smoothing processes.

**Strengths:**

- it proposes a novel label smoothing technique that integrates image information into the process, which is valuable for dense prediction problems like medical image segmentation.
- they conducted experiments on two publicly available segmentation benchmarks and compared their proposed method with existing soft-labeling strategies to evaluate its performance.
- The results showed that using GeoLS improved the segmentation accuracy over other methods, demonstrating its validity for integrating image information into label smoothing processes.

**Weaknesses:**

- The paper does not provide a detailed comparison of the proposed method with existing methods in terms of computational complexity.
- It is unclear how well the proposed method performs on datasets other than those used for evaluation.
- There is no discussion about potential applications or implications of this work beyond medical image segmentation tasks.

**Deanonymize Review:**

no

**Detailed Comments:**

- The paper is generally well written and I found no typos.
- Source code is provided, also during the review as anonymous repository, which is excellent.
- Statistical evaluation is sound.
- Label smoothing as presented here is a good idea to better capture class-wise relationships between pixels and spatial pixel-wise relationships captured in the geodesic distance transform, which includes more detail than traditional Euclidean distance between pixels.

**Paper Type:**

methodological development

**Questions To Address In The Rebuttal:**

- Provide a detailed comparison of the proposed method with existing methods in terms of computational complexity.
- Discuss potential applications or implications of this work beyond medical image segmentation tasks.
- the paper could be accepted as is but the system forces me to fill in more than 200 chars

---

### Official Review · Reviewer_GsLM · 2023-02-03

**Confidence:** 5
**Preliminary Rating:** 1
**Recommendation:** Poster

**Summary:**

This paper proposes a loss term, which includes the geodesic length of the (x,I(x)) surface. It is argued that such terms improve handling uncertainty in multi-class problems. The loss function is used for a 3d UNet and tested on 3 public domain (some) multi-modal datasets, and compared with 4 other loss functions from the literature. The new loss measure marginally outperforms most of the loss functions from the literature.

**Strengths:**

The paper focuses on the important aspect of the loss function in a standardized setting and on many publicly available datasets. It is well written and mostly easy to understand. It includes a statistical analysis of the performance of it as well as the competing losses.

**Weaknesses:**

The paper lacks a proper discussion on the surface of which the geodesic lengths are calculated. This may be deduced from (4), but is most likely not obvious to the average reader. Also, after reading the paper, it is still not clear to me why this geodesic loss function is the right model for a general set of problems. Further and from a physical perspective, the gamma term in (4) has the unit length/intensity and its values should thus be set in relation to the modality of x_i. A discussion on this is missing. I also find the claim, that each voxel-classification of 3D UNet does not take into account its neighboring classification in one-hot encoding since most of the layers share filters. Finally, the improvements are small

**Deanonymize Review:**

yes

**Detailed Comments:**

1. The illustration in Figure 2 are too small for me to appreciate the difference between training with the different loss functions.
2. Above (5), I don't understand what "along each voxel" means.


**Paper Type:**

methodological development

**Questions To Address In The Rebuttal:**

See above. See above. See above. See above. See above. See above. See above. See above. See above. See above. See above. See above. See above. See above. See above. See above. See above. See above. See above.

---

### Official Review · Reviewer_Hbq5 · 2023-02-03

**Confidence:** 4
**Preliminary Rating:** 5
**Recommendation:** Best Paper Award, Oral

**Summary:**

This article proposes a new soft labeling approach for learning-based image segmentation using geodesic distances, called Geodesic Label Smoothing (GeoLS). Contrary to existing soft labeling strategies, GeoLS includes image context (both pixel relationship and intensity) to better deal with the ambiguity in the boundaries between objects commonly found in medical applications.

In this approach, the hard-label resulting from the model are smoothed using geodesic maps computed for each class and a smoothing factor $\alpha$.

The authors evaluated their approach on two publicly available datasets, BraTS 2019 and FLARE 2021 and compared their results with three other soft labeling approaches (label smoothing, focal loss, spatially varying label smoothing) and the baseline one-hot encoded label. They also, evaluate the impact of the smoothing factor and the choice of the seed used to compute the geodesic maps.

**Strengths:**

- Very well written article with very clear context and motivation
- The proposed approach is interesting and tackles a real problem in image segmentation
- Extensive experiments (two different datasets, several compared methods, evaluation of the impact of the some method’s choices)

**Weaknesses:**

- The experiments could be more convincing if a cross-validation had been used instead of repeating the same experiment three times, in order to quantify the robustness to data variability.
- The approach seems computationally costly as many geodesic maps has to be computed. There is no discussion on this point nor any computational cost comparison with the other approaches.

**Deanonymize Review:**

no

**Detailed Comments:**

Nothing to report, this is a very good article !

**Paper Type:**

both

**Questions To Address In The Rebuttal:**

- Please discuss the computational cost of the proposed approach
- Please provide more details on the two seed generation strategies (with skeletonization and erosion). “Using remainings of skeleton and erosion operation of each target class” is not clear enough.
- In Eq 5 the multiplication is denoted $\star$ contrary to other equations. This should be corrected.

---

### Official Review · Reviewer_RSrG · 2023-02-04

**Confidence:** 3
**Preliminary Rating:** 4
**Recommendation:** Oral

**Summary:**

The authors address the problem of image segmentation and propose an approach to yield a better segmentation result. They propose smoothing the training labels using an approach that utilizes geodesic maps. Smoothing the labels has a positive effect on segmentation accuracy. The proposed approach is evaluated against a few other methods, and for some cases it obtains the superior result.

**Strengths:**

While wildly researched, segmentation is still one of the most important problems in image analysis and any approach for assisting and improving segmentation is valuable. The proposed method is simple yet it seems to be working well. The paper is rather well-written.

**Weaknesses:**

The paper describes a methodological development. Yet, the central parts of the methodology, covered in sections 2.2 and 2.3, seem unclear and/or self-contained. Furthermore, the reasoning and intuition behind geodesic label smoothing is not clear.

**Deanonymize Review:**

no

**Detailed Comments:**

The equation $g_c=$ in the last line of section 2.2 should also be placed centered in a line of its own, as it is an important step.

**Paper Type:**

methodological development

**Questions To Address In The Rebuttal:**

Please address the listed weaknesses. A question: What types of images is your method suited for? You test your method on medical images, is there a special reason for that? How do you expect your method to work on general segmentation problems? You may consider adding such reflection in your discussion section.

---

### Meta-Review · Area_Chair_49aW · 2023-02-25

**Recommendation:** Accept (Poster)
**Confidence:** 4

**Metareview:**

The authors propose  a new soft labeling approach for learning-based image segmentation using geodesic distances, called Geodesic Label Smoothing.  This method considers both class-wise relationships between labels and spatial pixel-wise relationships captured in the geodesic distance transform  to better deal with the ambiguity in the boundaries between objects commonly found in medical images. The methods is evaluated on two publicly available datasets, BraTS 2019 and FLARE 2021 and the results were compared with three other soft labeling approaches. Majority of the reviewers agreed that paper is well-written and the approach is interesting and experiments are sound. 	Some reviewers pointed out that performance improvements achieved with the proposed method is quite small. I agree with the majority of the reviewers and suggest acceptance of the manuscript.